# The Association of Parental Interest in Entrepreneurship with the Entrepreneurial Interest of Spanish Youth

**DOI:** 10.3390/ijerph17134744

**Published:** 2020-07-01

**Authors:** María-Isabel Luis-Rico, María-Camino Escolar-Llamazares, Tamara de la Torre-Cruz, Álvaro Herrero, Alfredo Jiménez, Pablo Arranz Val, Carmen Palmero-Cámara, Alfredo Jiménez-Eguizábal

**Affiliations:** 1Departamento de Ciencias de la Educación, Universidad de Burgos, Avda., Villadiego, n°1. 09001 Burgos, Spain; tdtorre@ubu.es (T.d.l.T.-C.); cpalmero@ubu.es (C.P.-C); ajea@ubu.es (A.J.-E.); 2Departamento de Ciencias de la Salud, Universidad de Burgos, Paseo Comendadores s/n, 09001 Burgos, Spain; 3Grupo de Inteligencia Computacional Aplicada (GICAP), Departamento de Ingeniería Informática, Escuela Politécnica Superior, Universidad de Burgos, Avda., Cantabria, s/n, 09006 Burgos, Spain; ahcosio@ubu.es; 4Department of Management, KEDGE Business, 680 Cours Libération, 33405 Talence, France; alfredo.jimenez@kedgebs.com; 5Departamento de Economía Aplicada, Universidad de Burgos, Plaza de la Infanta Doña Elena, s/n, 09001 Burgos, Spain; parranz@ubu.es

**Keywords:** entrepreneurship, entrepreneurial interest, youth, family, entrepreneurial eco-system, principal component analysis

## Abstract

As entrepreneurial interest is believed to represent a causal factor increasing entrepreneurship, research has begun to explore how family systems affect youth entrepreneurial interests. In the present study, we attempt to identify different types of family influence on the entrepreneurial interests of young people. A questionnaire was used to obtain data from 1633 Spanish adolescents (15 to 18 years old) and another questionnaire was used to obtain data from 839 parents. Principal component analysis identified unique family types and revealed that they have differential associations to entrepreneurial interest among youth. These findings reaffirm the influence of family on the entrepreneurial ecosystem and the promotion of an entrepreneurial family culture. This study further suggests that early attention should focus on the detection of entrepreneurial interest among youth so that actions can be implemented in the families to incentivize an entrepreneurial family culture.

## 1. Introduction

The past few years have seen increasing attention given to promoting entrepreneurship evident in interdisciplinary programs [1,2] and increasing social scientific research [3,4,5]. For example, in the case of Spain, national policies have encouraged entrepreneurship among schoolchildren and young people with programs such as Entrepreneurship in My School and Young European Company, aimed, respectively, at Primary Education and Compulsory Secondary Education [6]. Yet, and despite being one of the countries most affected by the financial crisis, Spain remains one of the developed countries with the lowest entrepreneurial rates. Although entrepreneurship cannot be thought to be the only solution for current social and economies woes, it is a relevant topic that affects almost 500 million people every year and is related to the creation of new enterprises [7], which has been identified as one of the key components for growth and social and economic development [8,9,10,11]. Additionally, entrepreneurship represents an increasingly accessible alternative work for young people who are usually those most affected by the crisis and imbalance of the economic system [12,13].

Considering these trends, it is hardly surprising that entrepreneurship has awakened unprecedented interest in the field of economics. Various economic variables have been employed to study the impact of entrepreneurship on the growth of productivity, competitiveness, economic growth, job creation, and even self-fulfillment [14]. In addition, policies of international scope that have centered on measures to stimulate entrepreneurial activity, the removal of barriers through grants, and the simplification of administrative procedures are yet to achieve the expected results [15,16,17]. This is demonstrated by studies such as Lopez et al. [18] who lament the absence of solutions from public initiatives to problems of Spanish rural entrepreneurship, especially in very depopulated areas. The deterministic conception of international policies lacks the pluralist and interdisciplinary views required to foster entrepreneurial interest, intention, and decision-making, especially among the young population, to move forward with the task of business creation and to understand the conditions affecting the situation and the timeframe [15,19,20].

For this reason, scholars have also pointed out that, in addition to the economic effects of entrepreneurship, it is also critical to analyze its implicit risks, as well as socio-educational aspects [21]. In particular, it is important to analyze the effect of relational aspects, such as the instrumental and emotional family supports [22], that are often determining factors of entrepreneurial interest [23,24]. Precisely, in this study we focus on the entrepreneurial interest of young people because it is a previous and under-researched step to entrepreneurial intention, which precedes and leads to entrepreneurial action. Specifically, our study, therefore, aims to make a contribution by analyzing whether family interest and perceived support are associated with the entrepreneurial interest of Spanish youth. Our findings highlight the importance of public policies in order to incentivize entrepreneurial culture.

## 2. Literature Review

Before reviewing the scientific literature that has studied family (relational) factors and their influence on the entrepreneurial interest of young people, we consider it important to introduce, at a conceptual level, the construct of entrepreneurial interest and its relationship with entrepreneurial intention. A person's interest in entrepreneurship is a previous and necessary step of entrepreneurial intention which, in turn, is a critical antecedent of the final conduct of entrepreneurship [8,10,25].

The social cognitive career theory (SCCT) of Lent et al. [26,27] explains the mechanisms through which individuals exercise control over the behaviors involved in the development of their careers. This is especially useful in explaining the initial phases (interest) of career decisions and the vocational behavior of adolescents and young adults immersed in their preparations for access to the world of work [28]. According to SCCT, the development of career goals (career choice process) is conceived as a sequence of interests, intentions, and behaviors that are presented as occupational choices where career intentions arise from the previous training of vocational interests [26]. The interests stimulate the career intentions or choice of goals (the plans of dedicating oneself to a certain activity and to achieve a certain result in the future), which increase the probability of making a career choice behavior (specific actions that mark the entry of the individual into an academic or professional line of work) [12,26,29]. From this perspective, interests are defined as early patterns of attraction towards occupational-related activities [30], career intentions or set-choice goals are plans to dedicate oneself to a certain activity, and career choice behaviors are specific actions that mark the entry of the individual into a professional career [28].

The construct of entrepreneurial interest and its consequent impact on the construction of employment projection has typically been studied in adolescence [31,32], a time when the consolidation of professional interests is considered to start [12]. In Spain, Lanero et al. [28] identified a positive and direct effect of entrepreneurial interest in university students on entrepreneurial intention, which in turn positively mediates with the final behavior of becoming an entrepreneur [12].

Consequently, it is important to study the entrepreneurial interest of young people, because, as already mentioned, it is a previous step to entrepreneurial intention. As the theory of planned behavior (TPB) [31,33,34,35,36] emphasizes, human intentionality immediately precedes action, situating itself as a central factor for planned behavior [37,38,39].

Moreover, as Schröder et al. [40] point out, empirical studies based on TPB demonstrate that intentions are the single best predictor of any planned behavior, including entrepreneurship [41]. In this sense, some studies analyze the stability of adolescents’ career interests and obtain empirical evidence to support it [40,42]. For example, Falck et al. [43] demonstrated the stability of career interests from adolescence to adulthood. In their study, students who stated entrepreneurial intentions at the age of 16 were shown to have a significantly higher probability of being an entrepreneur at the age of 33 compared to students who had not indicated any entrepreneurial intentions. Moreover, Schmitt-Rodermund [44] showed a relationship between entrepreneurial interests at the age of 13 and business start-ups and other entrepreneurial activities some 20 years later. Given the empirical support for the predictive value of intentions with regard to later behavior and the stability of career interest, we assume, drawing on Schröder et al. [40], continuity in the intentions expressed as early as adolescence related to self-employment in adulthood and thus have a substantive impact on the long-term succession planning [40].

About family factors, previous research has shown that the social models provided by relational factors such as the family setting positively influenced both in intention and the development of a professional career through self-employment [22,45,46,47,48] (i.e., coming from a nuclear family with a business link means that the person is introduced little by little into the world of entrepreneurship [49,50,51,52,53,54]. For example, Fayolle and Gailly [55] pointed out how different investigations have shown the importance of social status, cultural norms, and the model of parents and close relatives, in the formation of entrepreneurial interest in the young.

The scientific literature points to the family as being one of the main relational factors capable of predicting the intention of creating a company [56,57,58]. Authors such as Heck and Rogoff [59] considered that at every stage of a venture, the family connection is fundamental. The sharing of resources, including social networks, between the family and business is a major influence on the ability of each to thrive. 

Aldrich and Cliff [60] underlined that people are not atomized decision-makers, but rather, are implicated in networks of social relations. In this sense, some authors have highlighted the effect that the perceived entrepreneurial environment, within the family, has on the students´ entrepreneurial interest [61,62]. Students who belong to families with a family business perceive a more positive social pressure, both from their immediate environment and from society, with respect to his/her interest in entrepreneurship [63]. 

Schröder et al. [40], consistent with career development research found that adolescents´ perception on parental job rewards and identification with the family firm related to their career choice intentions. This implies that family business owners should be aware of their role model function concerning the rewards and costs of running a family firm. For example, talking positively about the family business at home may spur the offspring’s interest in the family firm. As positive human development results from the interplay between contextual and individual factors, parents should offer a stimulating context that allows their offspring to discover and crystallize their abilities and interests [40].

In this line, Edlman [22] showed that both instrumental support (family financial capital) and emotional support of the family (role of family cohesiveness) are necessary to increase the entrepreneurial interest of young people. Likewise, Criaco [45] argues that perceived parents' performance in entrepreneurship (PPE) enhances the interest of their children becoming entrepreneurs, due to exposure mechanisms. Other authors such as Laspita [46], found that the intergenerational transmission of business interests within the family is complex and involves more than one generation. Furthermore, the impact of entrepreneurial parents and grandparents on the offspring is not the same in all families, nor in all countries. Influences are especially strong in collectivist cultures (with high collectivism in the group). Among other findings, these authors also point out that entrepreneurial fathers and mothers had different effects on their sons and daughters. Likewise, it considers it necessary to stimulate future research on the mechanisms that underlie how business interests are transmitted within families.

All this evidence has made authors like Heckf and Rogoff [59] wonder how the role of the family has been largely overlooked as a key factor in promoting the entrepreneurial interest of their children, despite permeating most business ventures, surrounding virtually every entrepreneur, and providing a greater source and origin of motivation, education, and values that are critical to entrepreneurs. Gibb Dyer [64] and Aldrich and Cliff [60] pointed out how most studies ignore the impact of factors such as the family on the performance of a company and in the fostering of the entrepreneurial interests of their children. 

In summary, mounting empirical evidence has suggested that families play an important role in the venture creation process and thus deserve greater consideration in entrepreneurship and, therefore, in the entrepreneurial interest of young people.

Accordingly, and responding to the gaps mentioned above in theoretical investigation, the objectives of this study are:

Objective 1. To describe the association between interest in entrepreneurship among Spanish youth and the family’s interest in their child becoming entrepreneurial.

Objective 2. To examine the association between interest in entrepreneurship among Spanish youth and the family support that the young person would receive if he or she decided to become an entrepreneur.

Objective 3. To analyze what kind of family support is more prevalent among Spanish youth with high entrepreneurial interest and with families with a high interest in their children being entrepreneurs.

## 3. Methods

We conducted a questionnaire-based transversal descriptive study of a population with a probabilistic sample as part of a project coordinated at the national level and developed by the universities of Barcelona, Burgos, Deusto, La Rioja, and Santiago de Compostela and the National University of Distance Education. The questionnaire (in the Spanish language) had two parts, one to be filled in by the students (Student Questionnaire) (Appendix A) and another to be filled in by the families of the students (Family Questionnaire) (Appendix A). In addition, the University Pablo de Olavide de Seville and the University of Valencia assisted with the data collection activities.

### 3.1. Sample and Procedure

The student population under study comprised of students 15 to 18 years old from non-compulsory secondary education who were enrolled at teaching centers in Spain. The survey for students was carried out through a stratified sampling in educational centers of eight autonomous regions (Andalusia, Basque Country, Catalonia, Castilla and Leon, Galicia, La Rioja, Madrid, and Valencian Community). Based on data from the Ministry of Education, Culture, and Sport, we targeted 1764 students to achieve a confidence level of 95% with an error margin of 2.3%. However, 131 questionnaires were considered null due to missing responses and therefore removed from the final sample. Consequently, the final sample of students was 1633 (92.57% response rate). Regarding parents, 1764 questionnaires were sent to the parents of the students, and we received 839 valid responses (47.56% response rate).

Simple random sampling was used, maintaining the proportionality for each of the autonomous regions and for each educational level (67% high school students, 32.7% students of medium-grade vocational training, and 0.3% students of basic vocational training). The sampling units were selected from sectors during the 2013/14 academic year by selecting educational centers from each autonomous region at random, considering two criteria: one rural center in each autonomous region and a proportion of one private-state-assisted center for every three public educational centers. The questionnaire was administered at each of the selected centers to the number of students needed to cover the sampling quota. This fieldwork occurred during March and June 2014. Permission was sought from both the General Director of Education in each autonomous region and the directors of the educational centers prior to the administration of the questionnaire, and the directors were informed of the rationale behind the research. Two duly trained researchers personally attended the administration of the questionnaire at each center with the view to follow a properly standardized protocol of action. The sample of students comprised 50.1% women (*n* = 885) and 49.9% men (*n* = 879). The average age of the participants was 17.6 years old (SD = 1.60), and 89.6% of the participants were of Spanish nationality (*n* = 1.581).

### 3.2. Variables and Instruments

Due to the different interests of each of the universities collaborating in the study, the questionnaire was divided into the following thematic sections: students, life at the educational center, free time, family life, health and quality of life, studies and the labor market in the future, and entrepreneurship.

Following some of the guidelines that Shepherd [65] points out for the development of a valid scale, a pilot test in eight autonomous regions was completed to validate the questionnaire, establishing the stratification of the final sample and its proportionality as criteria. The number of questionnaires amounted to 10% of the subsequent sample. The pilot survey and results were then evaluated by fourteen experts from seven Spanish universities, who approved the definitive version and rated it as highly reliable. Likewise, the reliability of this questionnaire was contrasted with previous studies published by other authors [12,66,67,68].

To meet the proposed objective, the student and family questionnaires consisted of sections relating to the social, educational, occupational, and entrepreneurial variables (see Table 1).

### 3.3. Data Analysis

Having selected the questions/variables included in the study, a code was used as an identifier of each datum, thereby yielding a data set with 1633 cases (valid student questionnaires) and 72 characteristics or measures (answers given to each of the questions or variables analyzed).

The dataset was analyzed by means of principal components analysis (PCA), which is a statistical method based on reducing the dimension of the data being analyzed (reducing the number of variables), for the analysis of the results from the questionnaire. In other words, having a set of data with multiple variables, the end purpose is to reduce the number of variables in an effort to minimize information loss, as far as possible, for the new data.

The new factors of principal components obtained through the redimensioning process are each the result of a linear combination of the original variables and will, in turn, be independent from each other.

This technique was initially developed by Pearson [69] at the start of the 20th century, then studied and developed by Hotelling [70] and, more recently, has been applied in many studies on entrepreneurial intention in students [11,61,62,71,72,73]. The technique has two functions; on the one hand, it permits the optimal projection of the observations in a general N-dimensional space onto a space of reduced dimensionality (the principal components are the first step in identifying the possible latent or non-observed variables that generate the data), and on the other hand, it permits the transformation of the original variables, which are generally correlated, into new uncorrelated variables. If we start with a set of multivariable data, the end-purpose is to redimension this data set so that it has a smaller number of variables in decreasing order of importance and with a minimum loss of information so that the resulting variables are a linear combination of the original variables and are independent of each other.

The orthogonal base that maximizes the variance of the data must be found in order to project data onto a space of reduced dimensionality. To do so, the projection has to be established with the maximum variance, which will correspond to the first vector of the base (the first principal component). Subsequently, the projection that contains the highest remaining variance will have to be determined, which will correspond to the second vector of the base (the second principal component) and so on successively. When the data are projected onto the first few principal components, this will reduce the dimensionality while accounting for as much in the data as possible.

According to Bishop [74], PCA may be described as a map of vectors, Xd in an N-dimensional space projected on vectors, Yd, in an M-dimensional space, where M <= N, while X may be represented as a linear combination of a set of N orthonormal vectors *W_i_*:
$$x=\overset{N}{\underset{i=1}{\sum }}y_{i}W_{i}$$

Orthonormality is satisfied in the *W_i_* vectors as follows:
$$W_{i}^{t}W_{j}=\delta_{ij}$$
where *δ_ij_* is the delta of Kronecker.

## 4. Results

Following the analysis of the variables based on the data obtained with the questionnaire, the PCA projection of the dataset is shown in Figure 1, where each point is the representation of each one of the 1633 valid cases (student questionnaires) with its 72 selected characteristics (answers given to each of the questions or variables). In the analysis of the projection, we identify four groups in which the cases are distributed according to proximity and neighborhood. Clusters 1.1 and 1.2 correspond to the cases furthest from the part with the highest density and groups 2.1 and 2.2 correspond to the densest clusters (see Figure 1).

Based on the proposed work objectives, we analyzed the average scores of youth entrepreneurial interest (EI), the family’s interest in the young becoming entrepreneurial (EIC), and family support (FS) for each of the clusters resulting from applying the PCA technique (see Table 2).

We established a Cartesian axis as a function of the results in which the projections yielded by the PCA are found (Figure 2). We then considered the number of subjects present in each of the established groupings and the average scores of the variables in use, the family interest in their child becoming entrepreneurial and the family support that would be given if the young person decided to become an entrepreneur, discriminating and dividing the group into a larger number of individuals. Therefore, we found that group 2.1 obtained high results for both EI, EIC, and FS, group 2.2 obtained high results for EI and low results for EIC and FS, and group 1.2 had low results for both EI, EIC, and FS.

Having arranged the groups as functions of the PCA projections and the scores attained for the variables EI, EIC, and FS, we completed the characterization of each group in accordance with the responses obtained to the questions that form the unit of analysis (Table 3). We collected the following results, shown in Table 2, from the first section of questions referring to the survey data on the student profile.

We show the PCA grouping percentages of both questionnaires in Table 3.

Continuing with the groupings resulting from the application of the PCA technique, we described the characteristics of each of them based on the aspects analyzed:

Group 1.1: Low entrepreneurial interest (EI); high family interest for the young entrepreneur (EIC), and high family support (FS).

From the student questionnaire, we see that this group includes mainly men (64%)(Q.1), with both parents have a higher level of education (57%)(Q.10.1) and the percentage of families where the mother is the entrepreneur is of 28%, 14 points higher than families where only the entrepreneur is the father (Q.10.2). Members of this group feel highly satisfied with their family life (Q.20). They do not perceive that studies or work will help them to improve their lives (Q.37). They do not show answers as regards the motivations (Q.42) to create a company or the difficulties (Q.43).

From the questionnaire applied to families, the most prevalent type of support from the families is moral support, with a score of 4.9 out of 5 (Q.18.2).

Group 1.2: Low entrepreneurial interest (EL); low family interest for the young entrepreneur (EIC), and low family support (FS).

From the student questionnaire, we see that this group includes mainly men (82%)(Q.1), with both parents having a higher level of education (64%)(Q.10.1) and the percentage of families where the father is the entrepreneur is 28%, 10 points higher than families where the only entrepreneur is the mother (Q.10.2). Members of this group are not very satisfied with their family life (Q.20), with the feature with the highest score being the one that refers to the creation of their own family, with an average score of 3.5 out of 5. The perception of how studies and work help in their life (Q.37) displays scores below 2.5 out of 5. The motivations (Q.42) to create a company that the respondents have selected to a greater extent are economic independence (43%) and to earn money (43%). The difficulties that are perceived in creating a company (Q.43) are fear of failure (25%), lack of money to start (25%), and risk (21%).

From the questionnaire applied to families, none of the categories related to the family support exceeds the score of 2.5 out of 5, the highest one being moral support with 1.6 (Q.18.2).

Regarding these groups 1.1 and 1.2, although the sample sizes were only 14 and 28, respectively, they do provide useful information. Specifically, these groups point to the need to consider family interest as a key aspect in shaping the entrepreneurial system of young people.

Group 2.1: High entrepreneurial interest (EI); high family interest for the young entrepreneur (EIC) and high family support (FS).

From the student questionnaire, we see that this group includes mainly women (57%) (Q.1), with both parents having a higher level of education (54%) (Q.10.1). It should also be noted that the percentage where only the mother has higher education (15%), is the largest of all groupings. In this grouping, the percentage of families where the mother is the entrepreneur is 25%, five percentage points higher than families where only the father is the entrepreneur (Q.10.2). Members of this group are highly satisfied with their family life (Q.20) in all aspects analyzed, ranking higher than 3.4 out of 5. In the perception of how studies help them (Q.42), the aspects related to being successful in life and finding a job obtain scores above 4 out of 5. Referring to how work helps, the aspects related to having money, feeling useful, and becoming independent give scores of above 4 out of 5. The motivations (Q.42) to create companies that were pointed out more often by the young people surveyed are: putting my ideas into practice (61%), earning money (57%), and having economic independence (56%). The difficulties that precede creating a company (Q.43) include a lack of money (76%) and a fear of failure (63%).

In the questionnaire applied to families, the most prevalent type of support from the families is moral support, with a score of 4.7 out of 5 (Q.18.2).

Group 2.2: High entrepreneurial interest (EI); low family interest for the young entrepreneur (EIC) and low family support (FS).

From the student questionnaire, we see that this group is relatively evenly split between men and women (52% men) (Q.1), with both parents with higher education (51%) (Q.10.1), where the families in which the entrepreneur is the father are 23%, five points higher than the families in which the only entrepreneur is the mother (Q.10.2). Members of this group are highly satisfied with their family life (Q.20) in all the aspects analyzed, ranking higher than 3 out of 5. In the perception of how studies and work help them (Q.42), students obtain scores above 4 out of 5 in all aspects analyzed. The motivations (Q.42) to create companies more often indicated by the young people surveyed are: putting my ideas into practice (57%), earning money (62%), and having economic independence (53%). The difficulties that arise when creating a company (Q.43) are a lack of money (74%), fear of failure (62%), and risk (51%).

From the questionnaire applied to families, none of the categories related to the family support exceeded the score of 2.5 out of 5, the highest ones being moral support and ideas with 1.1 (Q.18.2).

## 5. Discussion

The resulting PCA projections provide us with a distribution among the groups, where the determination of the groups’ characteristics is useful for decision making in the fields of economic, educational, and curricular policy. Decisions oriented towards increasing entrepreneurial intention and its transformation into entrepreneurial action necessarily have to consider the characteristics of the group in line with the suggestions of previous studies [4,35,75,76,77]. In earlier works, the usefulness of PCA has been shown in multiple fields, ranging from the economic analysis of political risk [78] and in the area of computing [79] to the psychological aspects of entrepreneurship [12]. We contribute by applying this technique to the relational analysis of entrepreneurship. PCA provides a visualization of sample distribution patterns in accordance with the variables, which is a fundamental reference for subsequent decision-making in aspects as crucial as the social and the educational influence of the family.

In relation to the first objective, to describe the association between interest in entrepreneurship among Spanish youth and the family’s interest in their child becoming entrepreneurial, the data showed that the young population has a medium interest in entrepreneurship. The results from the PCA demonstrate the association between the family interest and the distribution of the cases, which allows us to identify and describe the characteristics of distinct groups in order to facilitate decision making. In general, these results corroborate the literature, which has established the influence of the family as a network of local entrepreneurship [29,52,54,57,58,59,63,66,67].

On the second objective, regarding the relationship between the entrepreneurial interest of young people and family support that the young person would receive if he decided to become an entrepreneur, the results show that family support guides the distribution of the projections of the cases studied, as is the case with the family interest for the child becoming entrepreneurial. Both scores (interest and family support) are similar in the groupings made. These results are aligned with those of authors such as Heck and Rogoff [59], Aldridh and Cliff [60], and Gibb Dyer [64] regarding the influence of the family on the entrepreneurial interest of young people.

Regarding the third objective, to analyze what kind of family support is more prevalent among Spanish youth with high entrepreneurial interest and with families with a high interest in their children being entrepreneurs, the grouped projections in which there is a high entrepreneurial interest of young people, together with a high interest and family support, showed that the kind of support they would receive to a greater extent is of the moral kind, rather than in economic means or ideas.

Another interesting aspect to consider is the results obtained between parental education level and entrepreneur families, where it is observed that the percentages that fluctuate most in the groupings with high family interest and high entrepreneurial interest are those related to the level of studies of the mother and the fact that the mother is the entrepreneur of the family. That is, families in which the mothers are entrepreneurs and have a higher level of education show higher levels of interest and family support for their child to become an entrepreneur.

One of the limitations of this work is related to the cross-sectional nature of the study which prevents us from establishing causal relations between the variables under study. Also, our study covers only eight regions of Spain due to data availability. Therefore, in future investigations, longitudinal designs and a broader geographical scope are proposed in which evidence might be unearthed regarding the relations between the variables in the context of the effects that are identified. Another limitation is related to the single-item nature of the questionnaire. Despite the fact that the validity and reliability of the questionnaire were confirmed through a pilot test conducted in eight autonomous regions in Spain, and was evaluated by 14 experts belonging to seven Spanish universities, it would be advisable to corroborate the results employing other alternative measures. Likewise, another limitation of this study is related to the generalization of the results, given the small samples in groups 1.1 and 1.2. However, it is also important to note that the fact that certain groups identified in PCA visualizations contain small samples does not mean that they are useless. PCA and other EPP techniques are frequently applied to identify outliers (including cases of one single datum isolated from the other data groups). Yet, we acknowledge the limitation in our paper that groups 1.1 and 1.2. are too small to provide reliable information regarding between-group differences. Only a much large sample size than what is available could completely rule out that groups 1.1 and 1.2 represent noise, rather than real subpopulations, due to potential sampling error and measurement error. Finally, future studies could also employ additional techniques such as cluster analysis, linear regression, or latent class/profile analysis to enlarge the findings and address additional research questions.

## 6. Conclusions

In this study, we focus on the role of the family as a factor positively associated with entrepreneurial interest among Spanish youth. The evaluation of family interest had been given scant attention in the study of entrepreneurship until only a few years ago. The under-estimation of family interest suggests the advisability, in terms of practical applications, of lending greater attention to the early detection of incipient entrepreneurial interest among young people, and to attributing due importance to the interest of the family in the design of actions to stimulate entrepreneurial culture.

Overall, this study contributes to the growing trend in the studying of the literature, not only the economic but also the relational aspects related to entrepreneurship, highlighting the role of the family on entrepreneurial interest among the Spanish youth.

The study also provides practical implications for policy-makers, by emphasizing the importance of including the family environment when designing actions and instruments aimed to increase the entrepreneurial interest of young people as a first step to foster entrepreneurial intention and action. Furthermore, our study underlines that when designing programs to promote the entrepreneurial interest of families, both families with and without entrepreneurial members should be taken into consideration.

## Figures and Tables

**Figure 1 ijerph-17-04744-f001:**
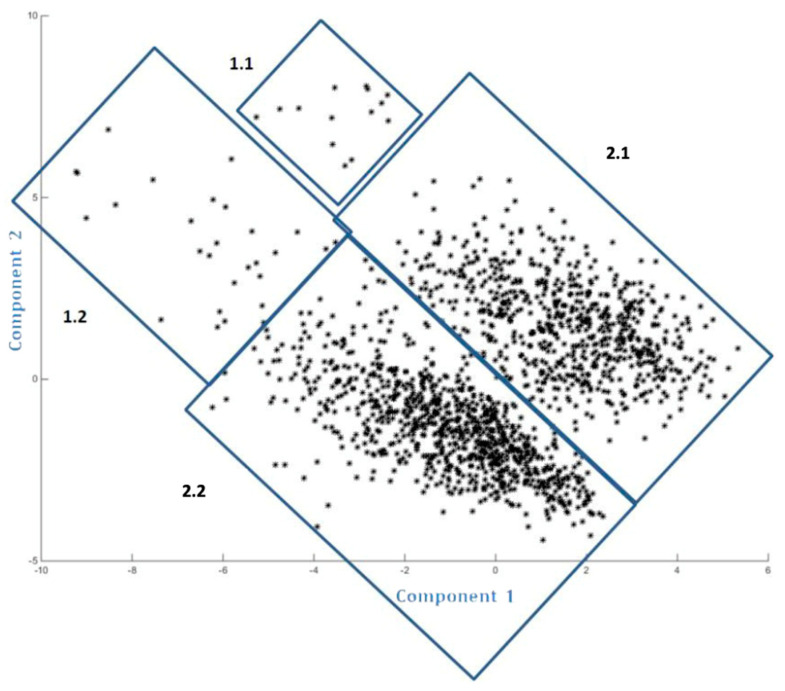
Results of the principal components analysis (PCA) analysis of the available data.

**Figure 2 ijerph-17-04744-f002:**
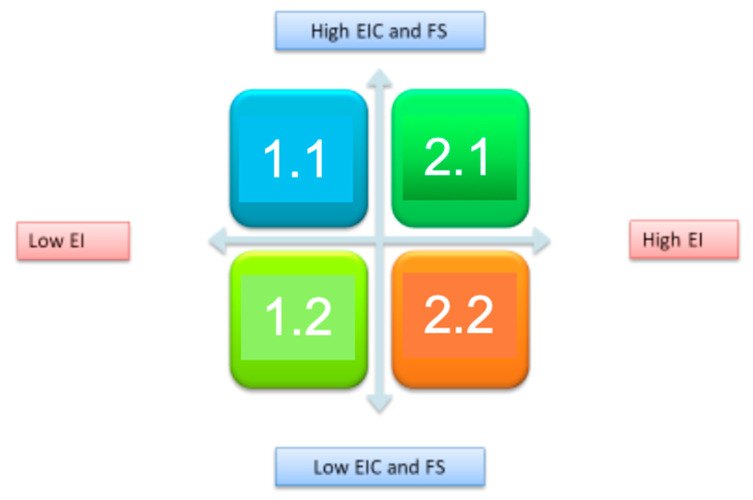
Representation of the average scores in EI, EIC, and family support (FS) in accordance with the groupings of the PCA projection.

**Table 1 ijerph-17-04744-t001:** Selection of questions/variables from the student questionnaire and the family questionnaire.

Respondents	Question Number and Type of Questionnaire	Section of Questionnaire	Data	Measures
Student	Q.1 Student questionnaire	Students	Sex	Woman/man
Student	Q.10 Student questionnaire	Q.10.1	Parental education	Only the father with higher education/both with higher education/only the mother with higher education/none with higher education
Student		Q.10.2	Entrepreneur families	Only the entrepreneur father/only the entrepreneur mother/both entrepreneurs/any entrepreneur
Student	Q.20 Student questionnaire	Family life	Satisfaction of the student with family life.	Likert Scale from 1 to 5
Student	Q.37 Student questionnaire	Studies and job market	Situations in which studying helps and aspects of situations in which working helps	Likert Scale from 1 to 5
Student	Q.40 Student questionnaire	Entrepreneurship	Degree of entrepreneurship interest (EI)	Likert Scale from 1 to 5
Student	Q.42 Student questionnaire	Entrepreneurship	Motives for creating business	Selection of the 3 most important reasons
Student	Q.43 Student questionnaire	Entrepreneurship	Difficulties for entrepreneurship	Selection of the 3 most important difficulties
Parent	Q.17 Family questionnaire	Entrepreneurship	Interest in your child becoming entrepreneurial (EIC)	Likert Scale from 1 to 5
Parent	Q.18 Family questionnaire	Q.18.1 Entrepreneurship	Family support (FS)	Likert Scale from 1 to 5
		Q.18.2 Entrepreneurship	Ways of helping your child as an entrepreneur: (1) economic support, (2) moral support, and (3) ideas	Likert Scale from 1 to 5

**Table 2 ijerph-17-04744-t002:** Average scores of entrepreneurial interest (EI) and family's interest in the young becoming entrepreneurial (EIC) and number of subjects per group.

Group	EI	EIC	Family Support	Number of Subjects
1.1	1	3.6	2.1	14
1.2	1.75	1.4	1.25	28
2.1	3	3.5	3.8	690
2.2 Null cases	2.9	1	1	901 131

**Table 3 ijerph-17-04744-t003:** Percentage breakdown by PCA group.

Type of Questionnaire	Question Number	Categories	Groups			
**Student Questionnaire**			**1.1**	**1.2**	**2.1**	**2.2**
	Q.1 Sex students by group	Woman	36%	18%	57%	48%
		Man	64%	82%	43%	52%
	Q.10.1 Parental education	Only father with higher education	14%	11%	10%	10%
		Both with higher education	57%	64%	54%	51%
		Only mother with higher education	0%	4%	15%	9%
		None with higher education	29%	21%	21%	30%
	Q.10.2 Entrepreneur families	Only entrepreneurial father	14%	28%	20%	23%
		Only entrepreneurial mother	28%	18%	25%	18%
		Both entrepreneurs	29%	36%	32%	35%
		None entrepreneur	29%	18%	23%	24%
	Q.20 Family life	Ideal FL	3.7	2.2	3.7	3.5
		Excellent FL	3.9	2.2	3.8	3.5
		Satisfaction with FL	4.2	2.6	4.1	3.9
		Achieve FL	3.5	2.1	3.7	3.5
		Change FL	3.7	1.8	3.4	3.2
		Own FL	4.6	3.5	4.4	4.3
	Q.37 Studies (S.) and Job (J.) market	S. help achieve success	1	1.6	4.2	4.1
		S. help to find work	1	1.6	4.2	4.2
		S. help with social relations	1	1.7	3.8	3.7
		S. help to earn money	1	1.1	3.8	3.8
		S. help though prefer to work	1	1.6	1.8	2.1
		J. helps to become independent	1	2.2	4.1	4.1
		J. helps family conviviality	1	1.5	3.8	3.9
		J. helps me to feel useful	1	1.7	4.1	4.1
		J. helps to earn money	1	2	4.4	4.4
		J. helps and is worth searching for	1	1.8	3.9	4
	Q.42 Motives form creating business	Family tradition	0%	0%	8%	9%
		Economic independence	0%	43%	56%	53%
		Social recognition	0%	7%	41%	36%
		Create employment	0%	7%	41%	36%
		Find work	0%	18%	32%	36%
		Direct	0%	7%	14%	13%
		Earn money	0%	43%	57%	62%
		Put ideas into practice	0%	18%	61%	57%
		Contribute to development	0%	14%	34%	32%
	Q.43 Difficulties of entrepreneurship	Risk	0%	21%	45%	51%
		Institute support	0%	14%	41%	32%
		Negative image	0%	4%	7%	8%
		Family support	0%	14%	4%	9%
		A lot of work	0%	4%	18%	20%
		Failure	0%	25%	63%	62%
		Bureaucracy	0%	14%	19%	19%
		Lack of money	0%	25%	76%	74%
		Training	0%	4%	17%	15%
		No ideas	0%	11%	21%	22%
**Family Questionnaire**	Q.18.2 Type of support for entrepreneurship	Economic Support	3.4	1.3	3.6	1
		Moral Support	4.9	1.6	4.7	1.1
		Ideas	4.3	1.5	4.3	1.1

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
