# Peer review of "The Association of Parental Interest in Entrepreneurship with the Entrepreneurial Interest of Spanish Youth"

_ijerph, 2020, doi:10.3390/ijerph17134744_

Round 1

Reviewer 1 Report

Dear Authors,

I appreciate the effort made by the authors to improve the manuscript “the association of parental interest in entrepreneurship to the entrepreneurial interest of Spanish youth.” Besides this improvement, I still see some problems with the manuscript.

  • The introduction does not position the manuscript clearly. You write about the entrepreneurial phenomenon in a very broad way, when reading the first two paragraph, one expects another type of manuscript more on the macro-level. We know the benefits of entrepreneurship, but why should we look at the entrepreneurial interest specifically? Why is this topic important?
  • This leads to my other point. What is the difference between entrepreneurial interest and the motivations of becoming entrepreneurs (or entrepreneurial intentions)? How do you define “entrepreneurial interest”? this definition is key for positioning your manuscript.
  • I am not an expert on Social Cognitive Career Theory but when I looked at the assumptions and the mechanism, I do not see it in your manuscript. The social cognitive career theory based on self-efficacy theory that represents that self-efficacy should be part of the model as an antecedent of entrepreneurial interest. Another antecedent of the career interest is expectations and learning experience. While learning experience is explained, entrepreneurial expectations are not developed in the arguments.
  • When you are developing your arguments, you should base on the literature on the topic. In other words, you attempt to understand students at the individual level, and then you base your argument on the firm level on page 3 (paragraphs 1 and 5). These arguments are not related to what you attempted to capture because you are studying students from 15-18 years, these adolescents are not close to becoming entrepreneurs or managers. I understand that the socialization process is key for developing interest in engaging in entrepreneurial activities but there is a long way for these students, you should consider their stage of life. A good example on how to do this is the paper of Schrörder et al. 2011, Schröder, E., Schmitt-Rodermund, E., & Arnaud, N. (2011). Career choice intentions of adolescents with a family business background. Family Business Review, 24(4), 305-321.
  • I would recommend showing the questions used during the survey. This is important because you have not based on previous scales and you need to show to the reader the questions from your questionnaire to give transparency to the method section. Once again, I have problems with the scales because several studies have captured these measures and you should have based on the prior scales while creating your questions.
  • A minor issue: Please check the English, you have some problems in the manuscript.

I hope that these comments are helpful. I hope that these comments are helpful.

Author Response

REVIEWER 1

Dear Authors,

I appreciate the effort made by the authors to improve the manuscript “the association of parental interest in entrepreneurship to the entrepreneurial interest of Spanish youth.” Besides this improvement, I still see some problems with the manuscript

RESPONSE: We are very glad to read that you find our manuscript improved and we honestly thank you for your additional feedback.

  • The introduction does not position the manuscript clearly. You write about the entrepreneurial phenomenon in a very broad way, when reading the first two paragraph, one expects another type of manuscript more on the macro-level. We know the benefits of entrepreneurship, but why should we look at the entrepreneurial interest specifically? Why is this topic important?

RESPONSE: Thank you for this comment. In this new version of the manuscript we have tried to emphasize the importance of entrepreneurial interest, both in the introduction and in the literature review, due its important role as a previous and under-researched step to entrepreneurial intention, which precedes and leads to entrepreneurial action.

  • This leads to my other point. What is the difference between entrepreneurial interest and the motivations of becoming entrepreneurs (or entrepreneurial intentions)? How do you define “entrepreneurial interest”? this definition is key for positioning your manuscript.

RESPONSE:  In order to clarify this point, we have introduced in the literature review the definition of entrepreneurial interest, and we have also clarified the difference between entrepreneurial interest and entrepreneurial intentions.

  • I am not an expert on Social Cognitive Career Theory but when I looked at the assumptions and the mechanism, I do not see it in your manuscript. The social cognitive career theory based on self-efficacy theory that represents that self-efficacy should be part of the model as an antecedent of entrepreneurial interest. Another antecedent of the career interest is expectations and learning experience. While learning experience is explained, entrepreneurial expectations are not developed in the arguments.

RESPONSE: Following your suggestion, in this revised version of the manuscript we have expanded the discussion on the Social Cognitive Career Theory, in order to explain the construct of entrepreneurial interest, and linking it to the Theory of Planned behavior.

  • When you are developing your arguments, you should base on the literature on the topic. In other words, you attempt to understand students at the individual level, and then you base your argument on the firm level on page 3 (paragraphs 1 and 5). These arguments are not related to what you attempted to capture because you are studying students from 15-18 years, these adolescents are not close to becoming entrepreneurs or managers. I understand that the socialization process is key for developing interest in engaging in entrepreneurial activities but there is a long way for these students, you should consider their stage of life. A good example on how to do this is the paper of Schrörder et al. 2011, Schröder, E., Schmitt-Rodermund, E., & Arnaud, N. (2011). Career choice intentions of adolescents with a family business background. Family Business Review, 24(4), 305-

RESPONSE:  Thank you for this insightful comment. Following your advice, in this revised version of our manuscript we have tried to avoid giving the impression that the paper builds on firm level arguments, and instead focused on the individual level. We also thank you for the very useful reference of Schröder et al. (2011), which we now include in the paper.

I would recommend showing the questions used during the survey. This is important because you have not based on previous scales and you need to show to the reader the questions from your questionnaire to give transparency to the method section. Once again, I have problems with the scales because several studies have captured these measures and you should have based on the prior scales while creating your questions.

RESPONSE: Thank you for your comment, we included the questionnaires in the previous version of our manuscript in the Supplementary Files S1 and S2. In this paper we employed a questionnaire validated by the Spanish Ministry of Education, Culture and Sport. This questionnaire has an official translation in the case of the Student Questionnaire, and both the Spanish and English translations are included in the new Supplementary Files S1 and S2. Unfortunately, there is no official English translation of the Family Questionnaire. We believe that attempting to translate it by ourselves could create distortions in the translation, especially given the short deadline to submit the revised version of the paper (only one week). Yet, we provide the Spanish version of the Family Questionnaire in Supplementary File S3, from which please note we only use questions Q17 y Q18 (Q18.1, Q18.2).

A minor issue: Please check the English, you have some problems in the manuscript.

RESPONSE: Thank you for pointing this issue. Although a professional copy-editor proof-read the paper in a previous round of review, we apologize for the remaining typos. Although we are not native speakers, we have carefully reviewed the text and corrected several errors, we hope the text looks better now. We hope no further typo remains, but if it does we will be glad to correct it if you kindly point it.

I hope that these comments are helpful. I hope that these comments are helpful.

RESPONSE: We truly thank you for your comments and suggestions during the review of the paper.

Reviewer 2 Report

The subject of this paper is of great interest given the importance of the topic of entrepreneurship and the pivotal role of families on the development of entrepreneurial interest, with interesting insights for governments looking for increasing entrepreneurship among young population. The paper has several strengths, but I would also recommend some changes to enhance the quality of the manuscript. The title and the abstract seems correct, as it is the main objective in the beginning of the article. The literature review seems also focused on the topic of study and includes recent publications in the field. Empirically, the authors rely on a double questionnaire and PCA as tool of analysis, which seem correctly employed, obtaining evidence of associations that can help entrepreneurship and educational-related public policy.

As suggestions to strengthen the paper, I recommend the authors the following:

  • I recommend the authors that, following standard practice in research, devote a specific section to the literature review, and not a sub-section of the introduction. I therefore recommend to name the literature review as section 2, and to move the numbering of the subsequent sections accordingly.
  • The sentence: “Considering these trends, it is hardly surprising that entrepreneurship as a multi-dimensional dynamic has awakened unprecedented interest in the field of economics.” I think you should drop the mention to “multi-dimensional dynamic” because you don´t focus on this, nor it is support by a reference.
  • Similarly, I recommend you to drop the sentence: “we continue to be partially unaware of the effectiveness of the association between the entrepreneurial initiatives and the final entrepreneurial behavior” because it is not clear what you mean by “initiatives”. Instead, you can simply link the first and second sentence of this paragraph as follows: “However, despite considerable scientific progress, a large part of the variation in the rates entrepreneurial activity across countries and sectors remains unexplained [12–16]”
  • In the literature review, please change “The investigation into entrepreneurial intention among students is still…” and replace by “Research on entrepreneurial intention among students is still…”. Also right after there is a paragraph which starts with “Finally”, but I am not sure why there is this finally, since there is no enumeration to finish. Consider maybe merging this paragraph with the previous one and changing the word “finally” or at least say “Finally, it is worth highlighting that…”
  • Also in this section, the paragraph starting with: “All this evidence has made authors like Heckf and Rogoff [56]…” is just a single sentence. Consider merging it with the previous or with the subsequent one as it makes more sense.
  • The expression “the young person would receive if he decided” please change it for the young person would receive if he or she decided.
  • Please simplify sentences such as: Based on the groupings obtained by the application of the PCA technique, we show the results obtained for each category analyzed both from the questionnaire of the young person and from the questionnaire of the families (Table 3)”. You can simply say “We show the PCA grouping percentages of both questionnaires in Table 3”. Similarly, please rename Table 3 as: Percentage breakdown by PCA group
  • When you explain the acronym (PPE], it is misplaced (it should go after the word “entrepreneurship), and also is starts with a parenthesis but ends with a bracket.

Author Response

REVIEWER 2

The subject of this paper is of great interest given the importance of the topic of entrepreneurship and the pivotal role of families on the development of entrepreneurial interest, with interesting insights for governments looking for increasing entrepreneurship among young population. The paper has several strengths, but I would also recommend some changes to enhance the quality of the manuscript. The title and the abstract seems correct, as it is the main objective in the beginning of the article. The literature review seems also focused on the topic of study and includes recent publications in the field. Empirically, the authors rely on a double questionnaire and PCA as tool of analysis, which seem correctly employed, obtaining evidence of associations that can help entrepreneurship and educational-related public policy.

RESPONSE: Thank you for your positive words toward our paper. We are very grateful for your constructive feedback, which has allowed us to improve the manuscript.

As suggestions to strengthen the paper, I recommend the authors the following:

  • I recommend the authors that, following standard practice in research, devote a specific section to the literature review, and not a sub-section of the introduction. I therefore recommend to name the literature review as section 2, and to move the numbering of the subsequent sections accordingly.

RESPONSE: Following your recommendation, we have made the literature review a section (section 2) in this new version of the paper and we have numbered the rest of sections accordingly.

  • The sentence: “Considering these trends, it is hardly surprising that entrepreneurship as a multi-dimensional dynamic has awakened unprecedented interest in the field of economics.” I think you should drop the mention to “multi-dimensional dynamic” because you don´t focus on this, nor it is support by a reference.

RESPONSE: We have dropped the unnecessary mention to “multi-dimensional dynamic”.

  • Similarly, I recommend you to drop the sentence: “we continue to be partially unaware of the effectiveness of the association between the entrepreneurial initiatives and the final entrepreneurial behavior” because it is not clear what you mean by “initiatives”. Instead, you can simply link the first and second sentence of this paragraph as follows: “However, despite considerable scientific progress, a large part of the variation in the rates entrepreneurial activity across countries and sectors remains unexplained [12–16]”

RESPONSE:  Thank you for this suggestion, we have followed your advice and linked both sentenced as you propose.

  • In the literature review, please change “The investigation into entrepreneurial intention among students is still…” and replace by “Research on entrepreneurial intention among students is still…”. Also right after there is a paragraph which starts with “Finally”, but I am not sure why there is this finally, since there is no enumeration to finish. Consider maybe merging this paragraph with the previous one and changing the word “finally” or at least say “Finally, it is worth highlighting that…”

RESPONSE:  Thank you very much for your suggestion, following the feedback from Reviewer 1, we have made some changes in the literature review in order to focus more on individual-level arguments, and as a consequence this sentence has been removed in the revised version  of the manuscript.

  • Also in this section, the paragraph starting with: “All this evidence has made authors like Heckf and Rogoff [56]…” is just a single sentence. Consider merging it with the previous or with the subsequent one as it makes more sense.

RESPONSE:  Thank you very much for your suggestion, we have merged it with the following paragraph.

  • The expression “the young person would receive if he decided” please change it for the young person would receive if he or she decided.

RESPONSE: Thank you very much for your suggestion, we have replaced the expression as you suggest.

  • Please simplify sentences such as: Based on the groupings obtained by the application of the PCA technique, we show the results obtained for each category analyzed both from the questionnaire of the young person and from the questionnaire of the families (Table 3)”. You can simply say “We show the PCA grouping percentages of both questionnaires in Table 3”. Similarly, please rename Table 3 as: Percentage breakdown by PCA group

RESPONSE: Thank you very much for your suggestions, we have followed your suggestion.

  • When you explain the acronym (PPE], it is misplaced (it should go after the word “entreprene urship), and also is starts with a parenthesis but ends with a bracket.

RESPONSE:  Thank you very much for your suggestion, we have correctly placed the acronym and replaced the bracket with a parenthesis.

Reviewer 3 Report

Thank you for the opportunity to revise this work. Overall, I think the paper meets the minimum quality standards to be considered for publication. It seems to be well balanced in terms of a theoretical framework that provides the context for empirical analysis. References are up to date and the use of PCA is innovative in the intersection between entrepreneurship and family support. The paper emphasizes the role of family support as a critical aspect for consideration in order to promote entrepreneurial culture, which has important theoretical and practical implications. Having said that, I have identified some parts of the paper that could be improved in my opinion: First, although the literature review seems comprehensive, I would recommend adding some specific references from the specific context of analysis (Spain), such as: Bernal Guerrero, A.; Cárdenas Gutiérrez, A.R. (2017). Evaluación del potencial emprendedor en escolares. Una investigación longitudinal. Educación XX1. 20, 73–94. López, M., Cazorla, A., Panta, M.P. (2019) Rural entrepreneurshipstrategies: Empiricalexperience in thenorthernsub-plateauofSpain. Sustainability 11(5),1243. In the second to last sentence of the introduction, you mention for the first time that your aim is to study the Spanish youth. You could provide an explanation about why the Spanish context is so important for you. For example, the significant impact of the financial crisis pronounced in Spain and the relatively lower rates of entrepreneurship that remain in this country compared to other countries nearby. You mention that your questionnaire covers 8 Autonomous regions in Spain. You could add a footnote specifying which ones, and also add as a limitation that you do not cover the whole country. I would recommend removing the mention to the funders and the (absence of ethical requirements) from the main text, and place it just in the section for funding (or as a footnote if it is really required by the funding agency). In the sentence: “These results respond to questions that authors as Heck and Rogoff [56], Aldridh and Cliff [16], and Gibb Dyer [15] posed, pointing to a clear influence of the family on the entrepreneurial interest of young people”, could you please clarify which questions you mean or rephrase the sentence? Also, is there a “such” missing before “as”? As a minor thing, please note that at times the distance between lines seems to be different throughout the paper, although this is something that is perhaps due to the software and can be corrected in the copy-editing phase of the paper, but still check if possible. Same for the alignment of the paragraphs on both sides.

Author Response

REVIEWER 3

Thank you for the opportunity to revise this work. Overall, I think the paper meets the minimum quality standards to be considered for publication. It seems to be well balanced in terms of a theoretical framework that provides the context for empirical analysis. References are up to date and the use of PCA is innovative in the intersection between entrepreneurship and family support. The paper emphasizes the role of family support as a critical aspect for consideration in order to promote entrepreneurial culture, which has important theoretical and practical implications. Having said that, I have identified some parts of the paper that could be improved in my opinion: First, although the literature review seems comprehensive, I would recommend adding some specific references from the specific context of analysis (Spain), such as:

Bernal Guerrero, A.; Cárdenas Gutiérrez, A.R. (2017). Evaluación del potencial emprendedor en escolares. Una investigación longitudinal. Educación XX1. 20, 73–94.

López, M., Cazorla, A., Panta, M.P. (2019) Rural entrepreneurshipstrategies: Empiricalexperience in thenorthernsub-plateauofSpain. Sustainability 11(5),1243.

RESPONSE: Thank you for your developmental feedback which has allowed us to improve the quality of our research. We are grateful for the suggested reference, which we now include in the paper.

In the second to last sentence of the introduction, you mention for the first time that your aim is to study the Spanish youth. You could provide an explanation about why the Spanish context is so important for you. For example, the significant impact of the financial crisis pronounced in Spain and the relatively lower rates of entrepreneurship that remain in this country compared to other countries nearby. You mention that your questionnaire covers 8 Autonomous regions in Spain. You could add a footnote specifying which ones, and also add as a limitation that you do not cover the whole country. I would recommend removing the mention to the funders and the (absence of ethical requirements) from the main text, and place it just in the section for funding (or as a footnote if it is really required by the funding agency). In the sentence: “These results respond to questions that authors as Heck and Rogoff [56], Aldridh and Cliff [16], and Gibb Dyer [15] posed, pointing to a clear influence of the family on the entrepreneurial interest of young people”, could you please clarify which questions you mean or rephrase the sentence? Also, is there a “such” missing before “as”? As a minor thing, please note that at times the distance between lines seems to be different throughout the paper, although this is something that is perhaps due to the software and can be corrected in the copy-editing phase of the paper, but still check if possible. Same for the alignment of the paragraphs on both sides. 

RESPONSE: Thank you for your detailed comments and suggestions. First of all, as you recommend, we have explained further the importance of the context in the introduction. We also mention the 8 Autonomous regions in Spain covered in our study (Andalusia, Basque Country, Catalonia, Castilla and Leon, Galicia, La Rioja, Madrid, and Valencian Community) and added this as a limitation of our study. We have also removed the redundancies in terms of funders, as this is included in the specific section for funding. We have also replaced the expression “respond to questions posed” as we agreed it was misleading, for the more correct “these results are aligned with those of”, and we have also added the missing “such”. Finally, we have homogenized the space between lines as well as the alignment of paragraphs.

This manuscript is a resubmission of an earlier submission. The following is a list of the peer review reports and author responses from that submission.

Round 1

Reviewer 1 Report

Dear Authors,

Thank you for giving the opportunity to revise the manuscript entitled „the association of parental interest in entrepreneurship to the entrepreneurial interest of Spanish youth”. This manuscript aims at understanding whether there is a relationship between family backgrounds perceived and entrepreneurial intentions of adolescents in Spain. The authors have collected a great database in different regions in Spain; I think that the research idea is interesting and could contribute to the entrepreneurship field. Besides these positive aspects, I have major concerns about the manuscript that I will summarize in the following lines:

  • The introduction is not effective. For example, in any moment you focus on what we know about the topic of the influence of parental role models in entrepreneurship, you do not mention the work of Criaco et al. 2017, Edelman et al. 2016, Laspita et al. 2012, to name few of the papers which have focus on the topic. Additionally, the topic of family interest and support comes by surprise in the research question, in other words, it is not well connected to the problem statement.
  • The introduction should provide to the reader “what we know”, “what we do not know” and “why it is important to study this”. So far, you do not answer these questions. I will recommend you reading Grant and Pollock (2011).
  • The section called “Review of scientific literature,” it should be called “Literature review” because you have a scientific study so your literature should be scientific only.
  • My major concern is in this section, I do not see the framework used and the authors should state what it is their position. On page 2, paragraph 5, the authors argued that they are two focus in the literature, entrepreneurial interest and family factors influencing entrepreneurial intentions. At that point, you should stay in which it is your direction and why you consider it is important to study. Then you will your arguments on these papers but you do not cite or work with the parental roles or family work.
  • Figure 1 is confusing and it has several problems. For example, in the theory of personality traits, the authors say in this box adaptability, trust in oneself, so on are theories of personality. Unfortunately, I do not think this is correct, I would say that a theory of personality would be the 5 big personality traits. Another example is sociological aspects and then you name demographic factors. So, I would recommend if you keep this figure, to eliminate the theory sublabel because it is not enhancing your explanation and creating problems.
  • You cite Sonnefelt et al. and I realize that two authors wrote the paper. Therefore, please cite in the manuscript “Sonnefelt and Kotter.”
  • Why do you have the objective on page 4? As you are doing quantitative work, I was expecting hypotheses, not objectives. I think that the authors should reconsider this and build some hypotheses instead of objectives in the back of the literature review.
  • The section called “Materials and methods” should be called “Methods” only.
  • In the method section, you need to clarify in which language you have done the survey. Did you merge the two samples? It would be important to show the scales used and who have used them before, or are these scales built by yourself? If yes, why did you do that?
  • I wondering if some of the questions that you asked in the survey are a good measure for your sample. For example, your sample is based on adolescents (young students from 15-18 years old) but you ask them questions such as difficulties of entrepreneurship. I am not sure if this is a good measure for such a sample, I think that you could ask their experience and perception of having entrepreneurial parents, such as quality time, negative or positive experience. I am not sure how aware these young people are about bureaucracy or training.
  • I do not underestimate the power of principal component analysis but you also need to use more methods to test the reliability and validity of your constructs. Why do not you use the Alpha Cronbach? I think that it will be useful to run a confirmatory factor analysis. I recommend reading the paper by Shepherd et al. (2009).
  • Table 2 need to be restructured, it is difficult to follow this table because it is on 3 different pages.
  • Why do you have group 1.1, group 1.2, group 3.1 and group 3.2, instead of group 2.1 and group 2.2?
  • I do not see how your manuscript contributes to the family business or business family literature, your study is mainly in entrepreneurship, specifically, Dyer 2002. Since your sample contains parents who are and are not entrepreneurs and you did not test for influence or causality, I think it is important that you clarify this.
  • Finally, revise your list of references. There are several issues, please follow the guidelines of the journal.

I wish you the best of luck!

Criaco, G., Sieger, P., Wennberg, K., Chirico, F., & Minola, T. (2017). Parents’ performance in entrepreneurship as a “double-edged sword” for the intergenerational transmission of entrepreneurship. Small Business Economics, 49(4), 841-864.

Edelman, L. F., Manolova, T., Shirokova, G., & Tsukanova, T. (2016). The impact of family support on young entrepreneurs' start-up activities. Journal of Business Venturing, 31(4), 428-448.

Grant, A. M., & Pollock, T. G. 2011. From the editors: Publishing in AMJ—Part 3: Setting the hook. Academy of Management Journal, 54(5), 873-879.

Laspita, S., Breugst, N., Heblich, S., & Patzelt, H. (2012). Intergenerational transmission of entrepreneurial intentions. Journal of Business Venturing, 27(4), 414-435.

Shepherd, D. A., Kuskova, V., & Patzelt, H. 2009. Measuring the values that underlie sustainable development: The development of a valid scale. Journal of Economic Psychology, 30(2), 246-256.

Reviewer 2 Report

The following corrections should be made:

(0) In lines 42-44, the meaning of the sentence is not clear

(1) In line 108, "start and" should be "start an"

(2) In line 120, "work on your own" should be "work on one's own" or "work on their own"

(3) In line 126, after "ability of each to thrive", the words "a venture's ability" should be removed

(4) In lines 146-147, "in fostering of entrepreneurial" should be either "in the fostering of entrepreneurial" or just "in fostering entrepreneurial"

(5) In line 152, from "who, when, and how some people" should just be "when and how some people" (i.e. remove "who")

(6) In line 154, "sum" should be "summary"

(7) In lines 161, 163 and 165, "Spanish youths" should be "Spanish youth"

(8) In lines 178-179, "The questionnaire for students were carried out" should be "The survey for students was carried out" (i.e. a questionnaire is not carried out; you carry out a survey using a questionnaire)

(9) In lines 186-187, the percentages don't add up to 100%

(10) In line 190, remove the "-" between "three" and "public"

(11) In line 213, "questionnaires were constructed of sections" should be "questionnaires consisted of sections"

(12) In line 225, "redimensioning the data under analysis" sounds awkward; replace it by "reducing the dimension of the data being analyzed"

(13) In lines 229-230, "are the result" should be "are each the result" (because each principal component is a linear combination)

(14) In line 234, "observations of a general" should be "observations in a general"

(15) In line 238, you have claimed that principal components "facilitate the interpretation of the data", but the general perception among statisticians is that principal components are usually difficult to interpret (i.e. losing interpretability is the price to pay for dimension reduction). So I think you should remove that claim

(16) In line 247, "onto the first principal" should be "onto the first few principal". In the same line, "these will" should be "this will"

(17) In line 248, "accompanied by as much" should be "while accounting for as much"

(18) In line 249, "Xd projected onto an N-dimensional space" should be "Xd in an N-dimensional space projected on"